# CdS-Based Hydrothermal Photocatalysts for Complete Reductive Dehalogenation of a Chlorinated Propionic Acid in Water by Visible Light

**DOI:** 10.3390/nano14070579

**Published:** 2024-03-26

**Authors:** Martina Milani, Michele Mazzanti, Claudia Stevanin, Tatiana Chenet, Giuliana Magnacca, Luisa Pasti, Alessandra Molinari

**Affiliations:** 1Dipartimento di Scienze Chimiche, Farmaceutiche ed Agrarie, Università di Ferrara, Via Luigi Borsari 46, 44121 Ferrara, Italy; michele.mazzanti@unife.it (M.M.); alessandra.molinari@unife.it (A.M.); 2Dipartimento di Scienze dell’Ambiente e della Prevenzione, Università di Ferrara, Corso Ercole I d’Este 32, 44121 Ferrara, Italy; claudia.stevanin@unife.it (C.S.); tatiana.chenet@unife.it (T.C.); 3Dipartimento di Chimica, Università di Torino, Via P. Giuria 7, 10125 Torino, Italy; giuliana.magnacca@unito.it

**Keywords:** chlorinated disinfection by products, photocatalysis, cadmium sulfide, visible light, dalapon, hydrodehalogenation reaction

## Abstract

Cadmium sulfide (CdS)-based photocatalysts are prepared following a hydrothermal procedure (with CdCl_2_ and thiourea as precursors). The HydroThermal material annealed (CdS-HTa) is crystalline with a band gap of 2.31 eV. Photoelectrochemical investigation indicates a very reducing photo-potential of −0.9 V, which is very similar to that of commercial CdS. CdS-HTa, albeit having similar reducing properties, is more active than commercial CdS in the reductive dehalogenation of 2,2-dichloropropionic acid (dalapon) to propionic acid. Spectroscopic, electro-, and photoelectrochemical investigation show that photocatalytic properties of CdS are correlated to its electronic structure. The reductive dehalogenation of dalapon has a double significance: on one hand, it represents a demanding reductive process for a photocatalyst, and on the other hand, it has a peculiar interest in water treatment because dalapon can be considered a representative molecule of persistent organic pollutants and is one of the most important disinfection by products, whose removal from the water is the final obstacle to its complete reuse. HPLC-MS investigation points out that complete disappearance of dalapon passes through 2-monochloropropionic acid and leads to propionic acid as the final product. CdS-HTa requires very mild working conditions (room temperature, atmospheric pressure, natural pH), and it is stable and recyclable without significant loss of activity.

## 1. Introduction

Photocatalysis by semiconducting oxides and chalcogenides is a technology of great importance because it can address both the problem of environmental pollution and that of clean energy shortage [1,2,3,4,5]. Many efforts have been carried out for photocatalytic degradation of toxic and hazardous pollutants [6,7]. However, when a new photoactive material is projected, it is important to have in mind the following issues: maximum absorption of excitation light, minimum charges recombination, and fast charge mobility towards the surface. These factors depend on the properties of the material, in terms of width of band gap, energy position of valence and conduction bands, crystal structure.

When the access to adequate quantities of water of acceptable quality is considered, one should realize that conventional water treatment plants use disinfection processes, which ensure drinking water safety but can generate disinfection byproducts (DBPs). In fact, these are formed by the reaction of disinfectants with many organic substances in water, such as organic natural matter, contaminants of emerging concern, pesticides, and halogenated compounds [8,9]. As a result, chlorinated substances often detected in treated water represent a kind of secondary pollution over time: their removal from the water is the final obstacle to its complete reuse [10,11,12].

In this regard, photocatalysis represents a promising methodology. Many research efforts have been carried out on advanced oxidation processes based on photocatalytic materials—TiO_2_, ZnO, and WO_3_ (with a narrow band gap)—that share high oxidizing capability and decrease DBP formation by oxidizing organic materials that might react with chlorine-based disinfectants. Nevertheless, hydroxyl radicals lead to the breakdown of carbon–carbon bonds, while carbon–halogen bonds remain intact. In addition, these processes can lead to the formation of dioxins as oxidation intermediates [13,14].

In contrast, research has made numerous advances in the reductive degradation of halogenated compounds, with formation of C-H bonds instead of C-X bonds (X being a halogen atom). Hydrodehalogenation has been studied with various catalytic systems, such as noble metals and H_2_ with alcohol as a hydrogen source, zerovalent iron catalysts, and metal cathodes; most of these require harsh reaction conditions (such as high temperature, high H_2_ pressure) [15,16,17,18].

In recent years, photocatalytic dehalogenation has attracted attention because the reaction could occur under mild conditions (at room temperature and atmospheric pressure) utilizing visible or solar energy [19]. The photocatalytic approach has shown promise, especially for the degradation of halogenated aromatic pollutants and polybrominated diphenyl ethers [20]. The used photocatalysts include semiconductor materials (metal oxides, mainly TiO_2_, metal chalcogenides) [21,22,23] or molecular sensitizers [24,25]. Yanagida and coworkers reported that several polychlorinated benzenes were dehalogenated using ZnS and CdS nanocrystals under UV and visible light irradiation, respectively, in the presence of triethanolamine as a sacrificial electron donor for the photogenerated holes [25]. Similar processes were studied also with CdS/TiO_2_ nanocomposites, exploiting the demonstrated improvement of charge separation efficiency [26,27,28,29]. Core shell ZnSe/CdS quantum dots had been used for hydrogenation or arylation of aryl bromides under 455 nm irradiation with DIPEA (N,N diisopropylethylamine) as a sacrificial donor [30].

However, when reductive approaches are used, aryl halides with electron donor groups are often not amenable to transformation, and compounds containing fewer halogens are more resistant to further reduction, making complete dehalogenation of aromatics very difficult to achieve [20]. Studies of hydrodehalogenation on other types of substances, such as aliphatic halides, are rare. For example, the reduction of a halothane (2-bromo-2-chloro-1,1,1-trifluoroethane) was carried out by photoexcitation with UV light of a platinized titanium dioxide (TiO_2_/Pt) [31]. The reaction led to the accumulation of bromide ions under aerated conditions, and the presence of noble metal was mandatory to detect some fluoride ions after an induction period.

In this work, we report about a synthetic CdS that, upon visible light excitation, is able to carry out the complete hydrodehalogenation of 2,2 dichloro propionic acid (dalapon) to propionic acid in an aqueous environment containing an alcohol as the sacrificial agent operating at room temperature and atmospheric pressure. The photocatalyst is prepared via hydrothermal procedure, and morphological, spectroscopic, and photoelectrochemical investigations are used to characterize it and to compare this material with both its unannealed parent and commercial CdS powder. The dehalogenation mechanism, clarified by HPLC–MS analysis, passes through the formation of 2-monochloro propionic acid and exploits electrons of the conduction band. ESR spin trapping experiments point out that alcohol is the hole scavenger and the source of needed hydrogen. A final treatment with oyster shells keeps cadmium ion concentration back within legal limits. The hydrothermal CdS shows good stability and recyclability. Interestingly, its capability in the reductive hydrodehalogenation of dalapon is compared to that of a previously synthesized hydrothermal CdS [32]: results evidence that preparation procedure and employed precursors have an important effect in determining electronic structure and photocatalytic properties.

To the best of our knowledge, this is the first contribution in which a stable, recyclable, visible light responding CdS-based photocatalyst can completely dehalogenate a chlorinated DBP, thereby decreasing the toxicity problem due to its presence in drinking water. Moreover, this work shows how similar materials but with significantly different properties can be obtained through control of the preparation method and paves the way for the development of a wide range of materials with adjustable reducing capacities.

## 2. Materials and Methods

### 2.1. Materials

Cadmium chloride (CdCl_2_, 99%) was purchased from Alpha Aesar (Karisruhe, Germany), thiourea (CH_4_N_2_S, 99%) from Aldrich (St. Louis, MO, USA) and 2-propanol (2-PrOH) from Fluka (Buchs, Switzerland). Mother solution of 2,2 dichloro-propionic acid (dalapon, 2000 ppm in MeOH) was purchased from Restek (Centre County, PA, USA). Acetonitrile (ACN), methanol (MeOH) and formic acid HPLC-MS grade from Merck (Sigma-Aldrich) (Darmstadt, Germany) were used for HPLC-MS analysis. For ICP-MS analysis, HNO_3_ 69% (Suprapur^®^) was purchased from Merck; HCl 37% (superpure) was purchased from Carlo Erba (Milan, Italy). Cadmium (Cd) Pure Standard, 1000 mg/L in 2% nitric acid (Sigma-Aldrich) was used for preparation of diluted standard solutions for calibration of ICP-MS. All the chemicals were used without further purification.

### 2.2. Preparation of Photocatalysts and Photoelectrodes

Hydrothermal synthesis was carried out following a procedure previously reported, albeit with some modifications [32]. Briefly, CdCl_2_ (5 mmol) and thiourea (10 mmol) were dissolved in water (40 mL) under magnetic stirring. After dissolution, the solution was transferred into a Teflon-lined stainless autoclave, and it was maintained at 160 °C for 10 h and then cooled at room temperature. The obtained powder was washed several times with water and ethanol and then dried. The resulting powder is labeled as CdS-HT. An amount of this powder was annealed in a muffle at 400 °C for 1 h and labeled as CdS-HTa. For electrochemical and photoelectrochemical measurements, fluorine-doped thin oxide (FTO) was used as ohmic support for the hydrothermal CdS materials. FTO was cleaned via sonication in 2-propanol for 10 min and dried under a warm air stream. It was added into the autoclave and the same experimental conditions described above for powder preparation were followed obtaining FTO/CdS-HT and FTO/CdS-HTa slides. Commercial CdS was deposited on FTO (FTO/CdS (commercial)) following the procedure already published in Ref. [32].

### 2.3. Morphology, Structure, and Composition of the Photocatalysts

Morphologies and compositions of the samples were analyzed by scanning electron microscopy (SEM, Zeiss EVO 40 scanning electron microscope) (Zeiss, Jena, Germany), equipped with an energy-dispersive spectroscopy EDS system (software AZTEC-Oxford, https://nano.oxinst.com/). Transmission electron microscopy (TEM) observations were performed using a TALOS L120C G2 instrument (Thermo Fisher, Waltham, MA, USA).

The phase structures of the as-synthesized samples were identified by X-ray diffraction using BRUKER D8 Advance X-ray diffractometer (Bruker, Billerica, MA, USA) equipped with a Sol-X detector, working at 40 kV and 40 mA. The diffraction patterns were collected in the 2θ range of 15–80° using an incident grazing angle set-up in a step-scanning mode with steps of Δ2θ = 0.02° and a counting time of 10 s/step using Cu Kα1 radiation (λ = 1.54056 Å).

### 2.4. Steady State Optical Absorption Measurements

The UV–visible diffuse reflectance spectra (DRS) of CdS-HT, CdS-HTa and commercial CdS powders were recorded with a Jasco V-570 spectrophotometer (JASCO, Tokio, Japan) equipped with an integrating sphere in the range of 200–800 nm and using BaSO_4_ as reference. The Tauc model was applied to evaluate the band gap energy of the catalysts. Tauc plots were obtained according to (F(R) × hν)^α^ = A × (hν − E_g_) where α = 2 for a direct band gap, (F (R) = (1 − R)^2^/2 R) is the Kubelka–Munk function, A is a proportionality coefficient, and E_g_ is the semiconductor band gap.

### 2.5. Electro- and Photoelectrochemical Investigation

Photoelectrochemistry characterization of FTO/CdS-HT, FTO/CdS-HTa, and FTO/CdS (commercial) slides was carried out using Metrohm Autolab PGSTAT 302/N (Methrom Autolab, Utrecht, The Netherlands) electrochemical workstation under solar simulated illumination with an ABET Sun Simulator (AM 1.5 G filter); the incident irradiance was set to 0.1 W/cm^−2^ with a Newport Power Meter model 1918-c. The experiments were performed in a three-electrode cell using aqueous HCOONa (1 M) as the supporting electrolyte; the reference electrode was SCE, the counter electrode was a Pt wire and FTO/CdS-HT, FTO/CdS-HTa or FTO/CdS (commercial) were used as working electrodes. Cyclic voltammetry (CV) measurements were performed between +1 and −1 V vs. SCE at a scan rate of 20 mV/s both under irradiation and dark conditions. For open circuit chronopotentiometry, the prepared thin films were positively polarized in the dark at +0.5 V vs. SCE for 100 s until a stable potential was reached. The film was then irradiated, generating electron–hole pairs that may undergo recombination or separation and storage in the semiconductor. The irradiation was maintained until a stable photo-potential was reached. Subsequently, restoration of dark conditions causes the decay of photo-potential due to recombination. Cyclic voltammetry experiments were carried out using glassy carbon or FTO/CdS HTa as working electrodes in acetonitrile solution containing LiClO_4_ (0.1 M) and dalapon (1.4 mM). The scan was between −2 V and +2 V, and the scan rate was 0.1 V/s vs. SCE. Acetonitrile was chosen as a solvent because it does not undergo reductive processes in the applied potential range.

### 2.6. Photocatalytic Experiments

In a typical experiment, selected photocatalyst powder (10 mg) was dispersed in an aqueous solution (3 mL) containing 2-PrOH (10% v/V) and dalapon (10 ppm). pH was adjusted to 7 via the addition of NaOH. Then, the solution was degassed by bubbling N_2_ for 30 min and irradiated for the desired period by an Oriel Xe/Hg lamp equipped with a glass cutoff filter (λ > 400 nm). Measured irradiation power is 150 mW/cm^2^. When the irradiation was stopped, the suspension was centrifuged (14.000 rpm, twice, 10 min each) and the supernatant was recovered for further analysis. For the recycling experiments, the powder was recovered after centrifugation, washed several times with deionized water, and employed again in a subsequent photocatalytic experiment under the same conditions. Removal of cadmium ions from irradiated solutions was accomplished by the addition of powder oyster shells (8 mg) to the solution. After 30 min of contact time at a controlled temperature of 38.4 ± 0.5 °C, the adsorbent was separated from the solution by filtration using 25 mm syringe filters with PVDF membrane 0.45 µm (Agilent Technologies, Santa Clara, CA, USA). The concentration of Cd^2+^ in the solution, before and after the contact with the adsorbent material, was determined by ICP-MS.

### 2.7. HPLC-MS Analysis

The concentrations of dalapon and its dehalogenated byproducts were determined by HPLC-MS analysis. The HPLC-MS system was equipped with an electrospray ionization (ESI) ion source. The mobile phase was obtained as a mixture of A (acetonitrile (ACN) and formic acid (0.1% v/V)) and B (water and formic acid (0.1% v/V)). Chromatographic separation was performed under gradient elution condition and the flow rate was 150 μL/min. The column (Supelco, Sigma-Aldrich, 150 mm × 2.1 mm) was packed with C18 silica-based stationary phase with a particle diameter of 2.1 μm. The injection volume was 2 μL for all standards and samples [33]. The employed gradient was: 0–1 min 5% A, from 1 to 12 min 5–95% A, then held isocratically at 95% of A for 3 min before reconditioning the column. To determine dalapon and mono-chloro propionic acid, ESI ion source was in negative mode, spray voltage 3 kV, capillary temperature 275 °C, capillary voltage −4 V, and tube lens −82.20 V. Propionic acid was determined in positive mode. MS experimental conditions were as follows: spray voltage 5 kV, capillary temperature 275 °C, capillary voltage 28 V and tube lens 50 V.

### 2.8. ESR Spin Trapping Experiments

Electron paramagnetic resonance (EPR) spin trapping experiments were carried out with a Bruker ER 200 MRD spectrometer equipped with a TE 201 resonator, at a microwave frequency of 9.4 GHz. Typically, powder of CdS (about 20 mg) was suspended in a 2-propanol solution containing α-phenyl N-tert-butyl nitrone (PBN, 5 × 10^−2^ M) as a spin trap and O_2_ as an electron acceptor. The samples were put into a flat quartz cell and irradiated (λ > 380 nm) directly in the EPR cavity. No signals were obtained in the dark during irradiation of the alcoholic solution in the absence of CdS.

### 2.9. ICP-MS Analysis

Cadmium concentration was determined by inductively coupled plasma mass spectrometry (ICP-MS); before analysis, samples were diluted with a solution containing HNO_3_ (1% v/V) and HCl (0.5% v/V), and all measurements were performed in triplicates. An Agilent 8800 Triple Quadrupole ICP-MS (Agilent Technologies, Santa Clara, CA, USA) equipped with a MicroMist glass concentric nebulizer, Peltier-cooled double-pass Scott-type spray chamber, and Ni cones was used for the analyses. The instrument-optimized operating parameters were 1550 W RF power, 8.0 mm sampling depth, 15 L/min plasma gas, and 1.03 L/min carrier gas. With the spray chamber temperature set at 2 °C, the isotopes measured were ^111^Cd, ^112^Cd, and ^114^Cd. The measurements were performed in No Gas mode, He mode and He-He mode, with a single-quad scan type. The He flow rates for the collision cell were 4.5 mL/min for the He mode, and 10 mL/min for the He-He mode. The integration time was 0.1 s for each mass value and the data acquisition was fixed at 3 replicates and 100 sweeps for replicates. Since all the isotopes show the same response using the different gas modes, the data presented in this work refer to the isotope 114 in the He mode.

## 3. Results and Discussion

### 3.1. Structural and Optical Properties

The XRD diffraction patterns of CdS-HTa (blue line) and CdS-HT (black line) are reported in Figure 1. Both show peaks (located at 2θ angles 24.8°, 25.5°, 28.2°, 36.7°, 43.8°, 47.9°, and 51.9°, respectively) that correspond to the (100), (002), (101), (102), (110), (103), and (112) planes. This pattern can be assigned to the hexagonal wurtzite CdS phase [34] by comparing it to the standard data from the JCPDS card (File No. 41-1049). After annealing, the intensity of the peaks increases and widths decrease simultaneously, indicating an increase of crystallinity upon thermal treatment. Annealing does not have any effect on crystalline phase composition of the material. In contrast, hydrothermal samples are quite different from the commercial CdS, whose XRD pattern is also reported in Figure 1 for easy comparison (red line): in fact, in addition to the hexagonal phase, commercial CdS shows the presence of cubic phase, which is characterized by a diffraction peak at 2θ ≃ 31°.

The average crystallite size (d) is calculated using the Debye–Scherrer formula (Equation (1)):(1)d=0.9λ/βcosθ
where d is the particle size, β is full width at half maximum (FWHM) in radians of the XRD peaks, θ is the diffraction angle, and λ (0.154 nm) is the wavelength of X-ray used. The calculated crystallite size was 27.3 nm for CdS-HT, and 35.6 nm for CdS-HTa showing that thermal treatment causes an increase in the size of crystalline domains. Concerning commercial CdS, a particle size of 48 nm was evaluated considering the diffraction peaks attributed exclusively to the hexagonal phase [32].

Figure 2a–d report the appearance and the SEM images of CdS-HT, CdS-HTa, and commercial CdS, respectively. The annealing changes the deep orange of CdS-HT into the golden yellow of CdS-HTa, while commercial CdS is orange in color as usual. SEM images show that CdS-HT has smaller assemblies than those of CdS-HTa; moreover, the annealed sample presents several large, flat-shaped aggregates (coin-like), which are not distinguishable in either the sample before annealing or in the commercial material. Considering that from XRD data we observe nanometric crystals and that from SEM images micrometric assemblies are visible, we can conclude that the particles observed by SEM are formed by aggregates of nanometric crystals.

Figure 2e reports the EDS pattern of CdS-HTa. Chemical analysis shows the presence of Cd and S, with an atomic ratio of about one, close to the nominal composition. A similar conclusion can also be achieved for CdS-HT (Appendix A) before annealing, indicating that the used synthetic procedure leads to a pure cadmium sulfide.

Figure 2f,g report two TEM images of the sample CdS-HTa. The thickness of the particles does not allow a deep investigation of the structural features of the material, as most parts of the particles are so thick as to avoid the transmission of the electron beam. Nevertheless, it appears clearly in Figure 2f that the synthesis originates a highly defective material, as several kinks and corners are visible in the more transparent parts of the particles. Although the presence of these defects, the crystallinity of the material is extended to the entire volume of the particles, as evidenced by the continuous interference fringe pattern visible in Figure 2g.

The optical properties of the materials were investigated using diffuse reflectance UV–Vis spectroscopy. From the reflectance results, Tauc plots were plotted for the calculation of the band gap energies. In Figure 3 the absorption spectra of CdS-HT and CdS-HTa are reported in KM units. Both samples show a light harvesting maximum efficiency in the range between 400–500 nm. Analysis of the Tauc plots (inserts of Figure 3, red dashed lines) shows that band gap values are in the range 2.2–2.3 eV. The band gap energy for commercial CdS was previously calculated to be 2.31 eV (Appendix A) [32].

### 3.2. Electro- and Photoelectrochemical Characterization

Dark cyclic voltammetry curves of FTO/CdS-HT and of FTO/CdS-HTa electrodes in aqueous solution containing HCOONa (1 M) recorded in the range (+0.2 V/−1.0 V vs. SCE) are reported in Figure 4 (black and blue, respectively). Neither of these electrodes show any visible wave in the range of applied potential. Only at strong negative polarizations, the reduction of electron acceptor states, probably conduction band states, is observed. According to a recent study [32], the absence of electrochemical waves at less negative potential values indicates that there are no intra band gap states acting as traps for electrons. This behavior is quite different from that exhibited by the electrode of commercial CdS on FTO (FTO/CdS), whose CV curve (red) is shown in Figure 4 for comparison: moving towards negative potential values, a not-intense wave is observable around −0.7 V. This peak and the corresponding return wave are attributable to the filling and subsequent emptying of the intraband trap states [32].

Figure 5 reports electrochemical experiments of potential vs. time using FTO/CdS-HT, FTO/CdS-HTa and FTO/CdS (commercial) as working electrodes in deaerated solutions of sodium formate (1 M). It is observed that when light is turned on after a dark equilibration time interval, the photovoltage of FTO/CdS-HTa (blue line, around −0.95/−1.0 V vs. SCE) is more negative than that of CdS-HT (black line, around −0.8 V vs. SCE). This result indicates that the reductive capability of the annealed CdS-HTa sample is higher than that of CdS-HT. In addition, CdS-HTa has a photopotential very similar to that of commercial CdS (red line), suggesting that these two materials should have comparable reducing power. When dark conditions are restored, the photo potential shifts back to the initial values quite rapidly for all samples, indicating that there are no intra-band gap states that could act as traps for electrons, thus decreasing their reducing capacity [32].

### 3.3. Photocatalytic Activity of CdS-HT and of CdS-HTa

The above characterization showed that the annealed CdS-HTa material is more crystalline and with a slightly larger band gap than CdS-HT, which decreases absorption in the visible region. Photoelectrochemical investigation revealed that CdS-HTa has a reducing capacity higher than CdS-HT and similar to that of commercial CdS. Therefore, it might be interesting to study the photocatalytic behavior of these materials in a process that requires a very high reducing capacity, such as a hydrodehalogenation process. The chosen molecule is 2,2 dichloro propionic acid (dalapon), a nonaromatic DBP, containing two chlorine atoms.

In a typical experiment, CdS-HTa or CdS-HT or commercial CdS (3 g/L) was suspended in an aqueous solution containing 2-propanol (10% v/V) and dalapon (10 ppm). Then, the suspension was degassed by N_2_ bubbling for 30 min. This time warrants the achievement of equilibrium conditions. Irradiation of the suspension (λ > 400 nm) followed. At the end of the illumination period, the powder was separated from the solution and the supernatant was analyzed by HPLC-MS. Figure 6 reports the decrease of dalapon during irradiation time.

It is observed that dalapon disappears completely during time upon visible illumination of CdS-HTa or commercial CdS. However, its disappearance is faster with the hydrothermal material since after 210 min irradiation more than 95% of dalapon was degraded. This result was achieved with commercial CdS only over a longer time. Despite its larger absorption of photons of the visible range, photoexcited CdS-HT is unable to cause any decrease in dalapon during time. These photocatalytic behaviors agree with the photoelectrochemical investigation (Figure 5), where CdS-HTa showed significant reducing power, very similar to that of commercial CdS and higher than that observed in the case of CdS-HT.

In addition, the decomposition of dalapon is accompanied by the formation of peaks, which were identified by MS as dechlorination intermediates, namely mono-chloropropionic acid and propionic acid, as can be seen in the chromatogram reported in Figure 7.

In Figure 8a,b, the time evolution of the relative concentration of 2-chloropropionic acid and propionic acid with respect to the initial concentration of dalapon are reported. It can be noted that concentrations of both the intermediates increase with time until dalapon undergoes decomposition. Subsequently, the concentrations of both the dehalogenated compounds decrease. Furthermore, 2-chloropropionic acid (Figure 8a) undergo a hydrodehalogenation reaction caused by the photoexcited chalcogenide and can be transformed into propionic acid.

Regarding propionic acid (Figure 8b), its concentration increases over time until approximately 250–300 min and then rapidly decreases. GC/MS analysis reveals the presence of ethyl propionate. This finding agrees with what has been reported in the literature, where ethyl propionate was detected in the photodegradation pathway of propionic acid by nanometric TiO_2_ (P25), most likely resulting from undegraded propionic acid and ethanol [35]. Although the more the aqueous solution of propionic acid is diluted, the more unidentified compounds are formed, Betts et al. followed the transformation of propionic acid (with an initial concentration of 63 ppm) into a mixture of identified CO_2_, ethanol, and acetic acid [36].

The formation of propionic acid from hydrodehalogenation of dalapon is a particularly relevant result because the toxicity of organic halides, which can be transformed in less halogenated compounds, is drastically decreased here.

The degradation of dalapon is a photocatalytic process in that the simultaneous involvement of a CdS-based photocatalyst and visible light are mandatory. In fact, irradiation (λ > 400 nm) of the aqueous solution of dalapon does not lead to any decrease in its concentration in the absence of CdS-HTa (or commercial CdS). Furthermore, no degradation occurs when the photocatalyst suspension is kept in the dark.

In a photocatalytic process, electrons promoted to the conduction band of CdS can be well exploited for dalapon dehalogenation if 2-propanol is an efficient hole scavenger. Since pathways involving holes generally entail the formation of radical intermediates, ESR spin trapping technique is a powerful tool for detecting radical species originating from 2-propanol photo-oxidation. For this reason, we carried out the following ESR spin trapping experiments: cadmium sulfide was suspended in the used solvent mixture containing α-phenyl N-tert-butyl nitrone (PBN) as a spin trap and irradiated directly into the ESR cavity. During irradiation, a spectrum consisting of a triplet of doublets (a_N_ = 13.87 G, a_H_ = 2.1 G) appears (Figure 9). No signals were observed in the absence of 2-propanol or in the dark. By comparison with previous literature [37,38,39,40] and considering the species present in the experiment, the obtained adduct is ascribed to the trapping of the iso-propoxy radical by PBN [40].

The transformation of dalapon into 2-chloropropionic acid and propionic acid is an example of photocatalytic dehalogenation reaction. Literature on electroreductive dehalo-genation on aliphatic chlorides pointed out that concerted dissociative electron transfer (DET) process is operative. It consists of simultaneous C-Cl bond breaking and free radical R^•^ associated with the release of Cl^−^ [17,41,42]. In those works, it was demonstrated that the electroreductive activation of C-Cl bond requires a catalytic step, and the nature of the electrocatalytic materials plays a crucial role in determining a remarkable positive shift of the reduction potential usually observed with respect to the inert electrode [17,41,42,43]. For this reason, FTO/CdS-HTa was used as a working electrode in cyclic voltammetry experiments in the presence of dalapon, and the voltammetric results were compared to a conventional glassy carbon electrode (Figure 10). CV of acetonitrile solution of dalapon (1.4 mM) and LiClO_4_ (0.1 M) as supporting electrolyte at the glassy carbon electrode shows that reduction of dalapon does not occur at potentials values less positive than −2 V (red line). When FTO/CdS-HTa electrode was used as working electrode, the beginning of a reductive process was observed around −1 V (blue line) and the irreversible wave was amplified (for equivalent overvoltage) with respect to the blank electrolyte (green line), suggesting the occurrence of an irreversible reduction process involving dalapon. In addition, when the same experiment was carried out in the absence of dalapon, the reductive processes involving the CdS display a cathodically shifted (by ca. 250 mV) onset potential. This result confirms that CdS has electrocatalytic properties towards the reduction of dalapon at a voltage (quasi-Fermi level) which is barely accessible upon illumination of this semiconductor in the presence of a good hole scavenger. This result also justifies the poor performance of CdS-HT, whose reducing capacity is lower than CdS-HTa (Figure 5 and Figure 6).

Experimental evidence from CV, ESR spin trapping, and chromatographic identification and evolution of the products allow us to propose the following reaction mechanism, schematized in Figure 1: excitation of CdS with visible light causes charges separation (Equation (2)). Holes are efficiently reduced by 2-propanol oxidation, which is also the source of protons.
(2)$$C\mathrm{dS}\overset{\;{}^{\circ}\mathrm{hv}\;(\lambda >400\mathrm{nm}){}^{\circ}\;}{\to }e_{CB}^{-}+h_{VB}^{+}$$

Then, electrons of the conduction band and protons are the leading players for dechlorination reaction, facilitated by the weakenign of C-Cl bond on the surface. The literature on electrocatalytic reduction of organic chlorides under aqueous conditions generally accepts the likely simultaneous occurrence of two mechanisms: direct electron transfer and indirect atomic hydrogen reduction (Ref. [41] and references therein). Considering that both of them lead to the same final dehalogenated products, it is beyond our interest to establish the relative weight of the two mechanisms.

When CdS-based photocatalysts are used, an often-occurring phenomenon is the photocorrosion of the material [44]. For example, commercial CdS is known to be susceptible to significant photodegradation and consequently has very poor photostability [32]. This characteristic severely limits its use in photocatalysis because the material is not reusable. In addition, during the photocorrosion process, cadmium ions—which have high toxicity—are released into the solution.

Within the overall effort of researchers to synthesize more stable and recyclable CdS-based photocatalysts [45,46,47], we evaluated the entity of photocorrosion and reuse of CdS-HTa. To investigate the stability of CdS-HTa, XRD analysis was performed on the photocatalyst before and after irradiation, and no evident structural modifications were observed (Appendix A). Therefore, CdS-HTa exhibits good structural stability. Additionally, further experiments demonstrate the reusability of CdS-HTa (Appendix A) for at least four cycles, as we observe a 15% decrease in photocatalytic conversion in the fourth recycle run. Nonetheless, the solution was analyzed using ICP/MS to determine the presence of Cd^2+^ ions and we found a concentration of roughly 10 mg/L (average of three repeated experiments) of Cd^2+^ ions, a value of the same order of magnitude as that among the best reported in the recent literature [46,47]. To overcome this drawback in the use of CdS, the solution resulting from the dalapon degradation was added with powder oyster shells (Appendix A). The concentration of cadmium in the filtered solution after 30 min contact time was 0.00432 mg/L, lower than the limit for drinking water [48].

### 3.4. Dependence of Photocatalytic Activity of CdS-HTa on Its Electronic Structure

CdS-HTa is a particularly attractive photocatalyst because it overcomes the main disadvantages of the commercial analogue without exhibiting a decrease in photocatalytic activity. In addition, it is a more photostable and recyclable material.

In addition, we compare two hydrothermal CdS samples: the first is the one described in this paper (CdS-HTa) and the second is a material prepared using cysteine as S source together with thiourea (CdS-HTa400), the properties of which have been recently reported [32].

These materials, which at a first glance should be very similar to each other, present quite different spectro-, electro-, and photoelectrochemical properties. In fact, Appendix A reports the absorption spectra of CdS-HTa and of CdS-HTa400: it is seen that extension of the absorption to longer wavelengths is more pronounced in the case of CdS-HTa400.

Open-circuit potentiometry experiments of FTO/CdS-HTa and FTO/CdS-HTa400 are compared in Appendix A: even though under illumination, the photovoltage is similar (around −0.9 V vs. SCE), when dark conditions are restored, the photopotential of CdS-HTa quickly shifts to the initial values, while photopotential decay of CdS-HTa400 is very slow. This behavior has been attributed to the presence of intra-band gap states (sulfur vacancies) that act as reservoir of electrons coming from the conduction band. The absence of intra-band gap states for CdS-HTa is confirmed also by dark cyclic voltammetry experiments, where no negative waves are observed in contrast to CdS-HTa400 for which it is evident a process at −0.6 V assigned to the filling of the trap states (Appendix A).

With all these results in mind, CdS-HTa400 absorbs more photons of the visible region and has a strong reductive capability due to improved charge separation. These properties seem to be useful premises for good photocatalytic activity. However, when CdS-HTa400 is employed for dalapon reductive dehalogenation reaction, no reaction occurs even after a long irradiation time. We believe that despite similar photopotential values, the electronic structure of the materials determines their photocatalytic behavior. Intra-band gap states collect electrons, thus slowing down the recombination of charges, but because their energy is lower than that of the conduction band, the accumulation of electrons in these states results in a decrease in the reducing capacity of the material, making it unable to carry out the hydrodehalogenation reaction. Therefore, small variations in the synthesis procedure of hydrothermal CdS can lead to different electronic structures, which consequently may result in different photocatalytic activities.

This intriguing behavior prompts us to study further to understand how the preparation of the material affects its electronic properties so that we can design the synthesis of a material with desired characteristics.

## 4. Conclusions

The present work describes a new hydrothermal CdS-based photocatalyst (CdS-HTa) with reducing properties similar to commercial CdS. Spectroscopic, electro- and photoelectrochemical investigation show that the electronic structure is important for the determination of the photocatalytic properties. In addition, experimental evidence of cyclic voltammetry shows that CdS-HTa has surface catalytic properties towards the reduction of chlorinated organic compound.

In fact, CdS-HTa photocatalyst is able to catalyze the complete reductive dehalogenation of 2,2 dichloro propionic acid (dalapon) from water using visible light. The reaction leads first to the formation of 2-chloropropionic acid and to propionic acid, thereby reducing the content of halogens and related toxicity. This achievement is of importance considering that dalapon is a disinfection by-product usually present in drinking water and its use is allowed in several non-crop applications.

The CdS-HTa operates in very mild conditions, such as room temperature and atmospheric pressure, and it is stable and reusable. Furthermore, the amount of Cd^2+^ still released during the photocatalytic experiment can be lowered below the law limit by treating the irradiated solution with powder oyster shells. These results demonstrate the potential for the development of a targeted strategy aimed at the preparation of photocatalytic materials with appropriate characteristics and provide an indication of feasibility from an application perspective.

## Data Availability

Data are contained within the article and Appendix A.

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
