# Peer review of "CdS-Based Hydrothermal Photocatalysts for Complete Reductive Dehalogenation of a Chlorinated Propionic Acid in Water by Visible Light"

_nanomaterials, 2024, doi:10.3390/nano14070579_

Round 1
Reviewer 1 Report
Comments and Suggestions for Authors
Martina Milani et al reported a paper entitled ‘CdS-based Hydrothermal Photocatalysts for Complete Reductive Dehalogenation of a Chlorinated Propionic Acid in Water by Visible Light’ where they described a new hydrothermal CdS-based photocatalyst with reducing properties similar to commercial CdS. They also investigated spectroscopic, electro-, and photoelectrochemical investigations showing that the electronic structure is important for the determination of the photocatalytic properties. The paper is interesting. Before publication, the author should address the following:
Ø What is the full form of HTa (CdS-HTa) please mention it in the abstract.
Ø Authors could cite a few references for the CdS-based similar type of studies in the introduction like (i) 10.1021/acsami.7b08407 (ii) 10.1021/acs.jpcc.8b02108 (iii) 10.1016/j.nanoen.2017.06.047
Ø In section 3.4. there is a spelling mistake in one line as Ttwo hydrothermal CdSsamples
Ø The authors should mention why the CdS-HTa sample shows a blue shift in absorption maximum than CdS-HTa. Although after annealing or growth time increase, generally, a red shift happens.
Ø Is it possible to take the TEM as well?
Ø The authors are encouraged to provide a schematic illustration of the proposed mechanism of photocatalysis.

Comments on the Quality of English Language
Minor editing of the English language required
Reviewer 2 Report
Comments and Suggestions for Authors
In this study, the authors employ a hydrothermal method to produce CdS-based photocatalysts for water purification. My main concern about this work is the use of CdS, which is one of the most dangerous materials. Indeed, currently, extensive research is devoted to avoiding such materials from the wast-scale application. However, in this study, the authors propose using this material for water purification, which might even bring another problem for long-term use. I don't suggest its publication.
Comments on the Quality of English Language
The English is fine.
Reviewer 3 Report
Comments and Suggestions for Authors
This paper reported the synthesis of CdS-based photocatalysts from hydrothermal reaction. The photocatalytic performance toward Reductive dehalogenation of chlorinated propionic acid was evaluated. The topic of the work was significant with regard to the development of photocatalysis technology. The manuscript can be considered for publication after the following comments are addressed.
Comments:
(1) The irradiation power used for photocatalytic experiments should be measured and provided.
(2) In Fig 6, why did CdS-HT show such almost negligible degradation efficiency? CdS-HT displayed noticeable reductive capacity as Fig 5 indicated. It was thus difficult to interpret the observation in Fig 6.
(3) In equations (2-5), the redox potentials should be provided, with which one can identify if the listed reactions can indeed occur from thermodynamics viewpoint.
(4) The band structure of the two CdS samples should be depicted in a scheme showing the energetic levels of conduction band, valence band and Fermi level.
(5) The chemical state and crystallographic structure of the samples upon recycling test should also be examined by XPS and XRD.
(5) Recent review articles summarizing the current challenge and future prospects of photocatalysis technology should be briefly introduced and cited to enlighten the readers: J. Phys. D: Appl. Phys., 2020, vol.53, pp.143001; Coordination Chemistry Reviews, 2021, vol.438, pp.213876.
